# Prevalence of ocular trauma and barriers to use of personal protective devices among welders in Hetauda, Nepal

**Sunil Thakali**[1]*, **Mohini Shrestha**[2], **Aleena Gauchan**[2], **Dikshya Bista**[2], **Hom Bahadur Gurung**[2]

**1** Hetauda Community Eye Hospital, Hetauda, Nepal, **2** Tilganga Institute of Ophthalmology, Kathmandu, Nepal

* sunilthakali@gmail.com

## Abstract

### Introduction

Welding poses significant ocular hazards in Nepal's industrial settings; however, data on trauma prevalence and protective barriers remain limited. This study assessed the burden of ocular trauma, personal protective equipment (PPE) practices, and associated risk factors among welders in Hetauda, Nepal.

### Methods

This cross-sectional study conducted in 2024 included 111 welders in Hetauda. Participants underwent comprehensive eye examinations and structured interviews. Logistic regression was used to assess factors associated with ocular trauma.

### Results

The prevalence of ocular trauma was 62.16%, primarily caused by metal chips (60.87%) and flames (37.84%). Although 78.39% of welders reported using PPE, 72.97% relied on non-certified sunglasses, and only 1.80% used certified protective goggles. Key barriers to appropriate PPE use included the absence of workplace mandates (62.50%) and a belief that protection was unnecessary (64.52%). Significant risk factors for trauma included male gender, lack of safety training (43.48%), and age between 46 and 60 years. Common ocular morbidities were corneal opacity (20.72%) and conjunctival congestion (10.36%).

### Conclusions

Despite reported PPE usage, the continued high rate of ocular trauma highlights deficiencies in equipment quality and training. Mandating certified eye protection, enforcing workplace safety policies, and integrating occupational health education are critical steps for preventing eye injuries in Nepal's industrial sector.

**Data availability statement:** All relevant data are within the manuscript and its Supporting information files.

**Funding:** This study was supported by Everest Parenterals, Nepal (https://everest-hcg.com/). Author ST (Sunil Thakali) received this funding. The funder had no role in study design, data collection and analysis, decision to publish, or preparation of the manuscript.

**Competing interests:** The authors have declared that no competing interests exist.

## Introduction

Ocular trauma is a major yet preventable public health issue, ranking third among occupational injuries after hand and foot trauma. Globally, around 2.5 million cases occur annually, with 500,000 leading to permanent blindness. Work-related injuries and illnesses contribute significantly to human suffering and economic loss worldwide [1,2].

International studies highlight the magnitude of this problem. In the United States, a narrative analysis of workers' compensation claims found that welders accounted for a substantial proportion of occupational eye injuries, most commonly caused by foreign bodies and burn, highlighting the critical role of consistent safety equipment use and worker training [3]. More recently, a large epidemiological study from United States documented a decline in welding-related ocular injuries between 2010 and 2019; however, it also demonstrated that mean aged 10–49 remained disproportionately affected, with a notable proportion of injuries occurring in non-occupational settings [4]. These reinforce the global relevance of welding related ocular hazards while highlighting the persistent gaps in prevention.

Welding, the predominant metal-joining method worldwide, is vital to metal fabrication but also a major cause of occupational injuries, especially in developing countries. Apprentices are exposed to toxic fumes, arc radiation, burns, and trauma, leading to both acute and chronic health issues. Welders face particularly high risks of eye injury due to mechanical, radiant, thermal, and chemical hazards [3,5,6]. While personal protective equipment (PPE) use is standard in industrialized countries, poor safety practices in small-scale industries across developing nations lead to preventable and often severe ocular trauma [7].

The International Labour Organization (ILO) promotes a preventive safety and health culture in the workplace to address the global burden of work-related accidents and illnesses, which cause severe human suffering and economic loss [2]. However, in Nepal's industrial hub Hetauda, systematic reporting of such incidents remains scarce. Data from Hetauda Community Eye Hospital reveal that welding and metalwork account for 17.7% (379 cases) of all eye injuries, the highest among occupational causes [8]. These findings highlight welding as a disproportionately high-risk occupation for eye injuries in Nepal's industrial sector.

While the dangers of welding are well-documented globally, critical gaps persist in Nepal's context, particularly in industrial areas like Hetauda. There is limited data on welders' awareness of occupational hazards, their adherence to personal protective equipment (PPE) use, and the barriers to implementing safety practices. This study aims to (1) assess welders' knowledge of workplace hazards, (2) identify obstacles to PPE utilization, and (3) evaluate existing safety measures. By addressing these gaps, the findings will inform targeted interventions to reduce preventable injuries and improve occupational health outcomes for Nepal's welding workforce.

## Materials and methods

### Ethical clearance

Ethical approval was obtained from the Institutional Review Committee (IRC) of Tilganga Institute of Ophthalmology (Ref. No. 07/2024) on April 22, 2024. The study

adhered to the principles of the Declaration of Helsinki. All participants were informed about the study objectives, procedures, potential risks, and benefits, and provided written informed consent. Patient privacy and confidentiality were strictly maintained throughout data collection, cleaning, and analysis. Participants were recruited only after receiving IRC approval and providing informed consent.

## Study design and sample size

This was a prospective cross-sectional observational study conducted among welders in Hetauda Municipality, Nepal, in 2024. Only registered welders from welding workshops within the city, as recommended by the Grill and Steel Fabrication Association of Makwanpur, were included. A list of these workshops was obtained from the association, and all identified welders were made to invite for participation-effectively attempting a census of the welding population in Hetauda.

The required sample size was calculated using the single-proportion formula:

$$n = Z^2 \cdot p \cdot (1 - p)/d^2$$

where $Z = 1.96$ for a 95% confidence interval, $p = 0.479$ (47.9%), based on a previous study in Accra, Ghana [1], and $d = 0.05$. The estimated sample size for an infinite population was 384. Applying finite population correction for an estimated 600 welders in Hetauda yielded an adjusted sample size of 235. After accounting for 10% non-response, the target sample size was 262.

The flowchart showing the stepwise recruitment of welders is presented in Fig 1. In practice, however, recruitment was limited to welders registered with the Grill and Steel Fabrication Association of Makwanpur, as only these workshops could be accessed through official records. Out of the registered welders, 111 were consented, forming a convenience sample, as participation was limited to accessible and willing individuals during the study period. This limitation is acknowledged when considering the generalizability of the findings.

Eligible welding workshops were identified in Hetauda Sub-Metropolitan City with the help of the Grill and Steel Fabrication Association of Makwanpur. All registered welders were invited to participate. A total of 111 welders provided informed consent and were included in the final analysis.

## Inclusion and exclusion criteria

All welders working in the registered workshops during the study period were eligible. Welders who refused participation or were unable to complete the questionnaire were excluded All enrolled participants underwent a complete eye examination irrespective of the presence or absence of ocular complaints. This approach ensured capture of both symptomatic and asymptomatic ocular trauma.

## Data collection tools

Data were collected using a structured questionnaire adapted from a validated study in Accra, Ghana [1], with minor modifications for cultural and occupational relevance in Nepal. These included items on local welding equipment, Nepali terms for protective devices, and adjustments to occupational history questions to reflect local work practices. The adapted questionnaire was reviewed by occupational health and ophthalmology experts to maintain content validity and was pilot-tested among a small group of welders (n = 10) to ensure clarity, comprehension, and contextual appropriateness before full-scale data collection.

The study questionnaire comprised five sections designed to capture relevant information from participants (Table 1). The Demographics section included questions on age, sex, years of experience, education, and type of workshop. The Knowledge of hazards section assessed awareness of ocular risks, types of injuries, and use of personal protective equipment (PPE). The PPE use and barriers section explored the frequency, type, accessibility, and

**Target Population: Registered Welders in Hetauda Sub Metropolitan City (as recommended by Grill and Steel Fabrication Association of Makwanpur)**

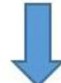

**Obtained List of Welding Workshops from the Association**

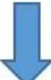

**Identified All Welders from the Registered Workshops**

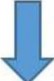

**Invited All Identified Welders to Participate (Attempted Census Approach)**

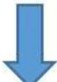

**Welders Who Gave Consent and Participated (n = 111)**

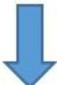

**Final Sample: Convenience Sample of available Welders**

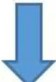

**Data Collection Conducted**

**Fig 1. Flowchart of participant recruitment.** Diagram showing the stepwise process used to identify registered welding workshops, invite welders for participation, obtain consent, and include eligible participants in the final analysis.

**Table 1. Components of the study questionnaire.**

| Section | Number of Questions | Description |
|---|---|---|
| Demographics | 5 | Age, sex, years of experience, education, workshop type |
| Knowledge of hazards | 8 | Awareness of ocular risks, types of injuries, PPE use |
| PPE use and barriers | 6 | Frequency, type, accessibility, affordability |
| History of ocular trauma | 5 | Self-reported or clinically confirmed injuries |
| Safety practices | 4 | Workplace protocols, training received |

The table summarizes the five sections of the questionnaire and the number of questions and key domains covered in each section.

affordability of protective measures. The History of ocular trauma section recorded self-reported or clinically confirmed injuries, while the Safety practices section evaluated workplace protocols and any training received. In the questionnaire, Frequent Safe Training refers to safety education or demonstrations on the use of protective devices conducted at least

once every six months, whereas Sometimes Safe Training refers to irregular or occasional safety instruction provided less than once per year.

### Ophthalmic examination

All participants underwent a comprehensive ocular examination. Visual acuity was assessed using a Snellen chart, followed by slit-lamp biomicroscopy to evaluate the anterior segment. Fundoscopy was performed using a 90D lens under slit-lamp examination, supplemented with indirect ophthalmoscopy. No posterior segment abnormalities, including maculopathy, were detected. Intraocular pressure (IOP) was measured for all welders using an iCARE non-contact tonometer.

### Definition of ocular trauma

Ocular trauma was defined as any self-reported or clinically confirmed injury to the eye or periocular structures from welding activities, including foreign bodies, corneal or conjunctival abrasions, chemical or thermal burns, and blunt trauma. Both past self-reported history and clinical finding during examination were considered when classifying ocular trauma

### Data analysis

Each participant underwent a complete eye examination and was interviewed using a structured questionnaire at Hetauda Community Eye Hospital. The questionnaire was adapted from a previous study conducted in Accra, Ghana, with minor modifications. Ocular trauma occurrence was based on self-reporting by participants. Data were cleaned and de-identified using Microsoft Excel (version 16.0). Statistical analysis was conducted using R version 2025.05.1 (R Core Team, Vienna, Austria). Categorical variables were presented as frequencies and percentages. Continuous variables were summarized as means with standard deviations(SD). Logistic regression was used to assess the association between various factors and ocular trauma. A p-value $< 0.05$ was considered statistically significant

## Results

### Participant characteristics

A total of 111 welders were invited to participate, and all consented, yielding a 100% response rate. The majority were male (n = 103, 92.79%), with a mean age of 32.12 ± 12.21 years (range: 17–67) and a median age of 29 years. Education levels varied: 9.01% had no formal education, 25.23% completed primary education, 43.24% secondary education, and 22.52% higher secondary (Table 2).

### Personal Protective Equipment (PPE) ownership and use

Of the 111 welders, 80 (72.07%) reported owning PPE, and 87 (78.39%) reported using it during work. Among PPE users, the majority used protective glasses (72.97%), while a small number used goggles (1.80%) or eye shields (3.60%). Among the 31 who did not own PPE, the most common reason was perceiving it as unnecessary (64.52%), followed by expense (22.58%) and lack of access (12.90%). For the 24 who did not use PPE despite owning it, the most cited reasons were: lack of mandate (62.50%), perceived low risk (16.67%), and discomfort (8.33%) as shown in Table 3. An illustration of the sunglasses commonly used by welders in Nepal is depicted in Fig 2.

### Prevalence and causes of ocular trauma

The prevalence of ocular trauma among the welders was 62.16% (n = 69, 95% CI: 52.90% to 70.60%). The leading cause of ocular trauma was flying metal chips (60.87%), followed by exposure to flames (37.84%) and other causes (1.45%).

An illustration of metal chips sticking on the cornea of welders is depicted in Fig 3, representing the most frequently encountered form of ocular trauma in this occupational group.

**Table 2.  Demographic characteristics of welders included in the study (N = 111).**

| Characteristics | N (%) or Mean ±SD |
|---|---|
| **Gender** | |
| Male | 103 (92.79%) |
| Female | 8 (7.21%) |
| **Age (years)** | |
| Mean age ± SD | 32.12 ± 12.21 |
| Median (Range) | 29 (17-67) |
| **Education level** | |
| No formal education | 10 (9.01%) |
| Primary (up to 5) | 28 (25.23%) |
| Secondary (up to SEE/SLC) | 48 (43.24%) |
| Higher secondary (+2) | 25 (22.52%) |

**N (%)** indicates number of participants and percentage of the total; **Mean ± SD** indicates the average and standard deviation; **SEE**, Secondary education examination; **SLC**, School leaving certificate.

**Table 3.  Personal Protective Equipment (PPE) ownership and use among welders (N = 111).**

| Characteristics | N (%) |
|---|---|
| **Own PPE (N = 111)** | |
| Yes | 80 (72.07%) |
| No | 31 (27.93%) |
| **If no, reason for not having PPE(n = 31)** | |
| Do not know where to get one | 4 (12.90%) |
| Not necessary | 20 (64.52%) |
| Too expensive | 7 (22.58%) |
| **Use of PPE(N = 111)** | |
| Yes | 87 (78.39%) |
| **Types of PPE used**[a] | |
| Glasses | 81 (72.97%) |
| Eye shields | 4 (3.60%) |
| Goggles | 2 (1.80%) |
| **No** | 24 (21.62%) |
| If no, reason for not using PPE (n = 24) | |
| Feels uncomfortable | 2 (8.33%) |
| Low risk at Task | 4 (16.67%) |
| Short duration of task | 3 (12.5%) |
| Use of eye PPE not mandatory | 15 (62.50%) |

**N (%)** indicates number of participants and percentage of the total.

[a]Multiple responses possible; percentages based on total respondents using PPE (N = 87).

## Characteristics of welders with ocular trauma

The mean age of welders with ocular trauma was 30.81 ± 12.30 years. The majority were male (97.10%), while only 2 of the 8 female participants (2.9%) experienced ocular trauma. Of these, 72.46% owned PPE and 79.71% reported using it. A majority (91.30%) had received training in PPE use, yet 43.48% had never undergone formal safety training. Notably, 81.16% were wearing protective eyewear at the time of injury (Table 4).

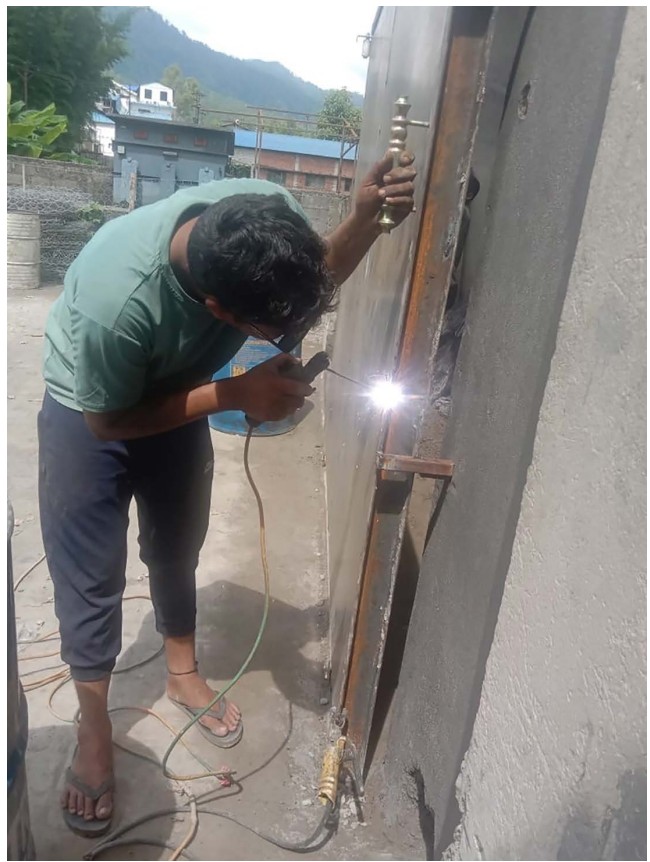

**Fig 2. Sunglasses commonly used by welders in Nepal.** An illustration showing the typical protective eyewear worn by welders in Nepal to reduce exposure to bright light, sparks, and ultraviolet radiation.

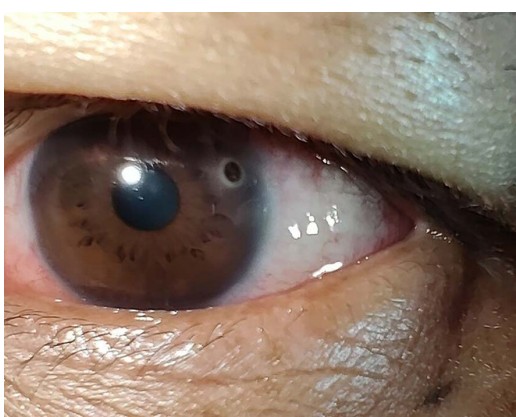

**Fig 3. Corneal foreign body following welding injury.** Corneal foreign body (metal chips) seen in a welder, representing the most commonly observed ocular trauma in this occupational group.

**Table 4. Characteristics of Ocular trauma patients among the welders (N = 69).**

| Characteristics | N (%)or Mean ±SD |
| --- | --- |
| **Age(years)** | |
| Mean age±SD | 30.81±12.30 |
| Range | 17-67 |
| Median age | 26 |
| **Gender** | |
| Male | 67 (97.10%) |
| **Education level** | |
| No formal education | 6 (8.70%) |
| Primary (up to 5) | 16 (23.19%) |
| Secondary (up to SEE/SLC) | 33 (47.83%) |
| Higher Secondary (+2) | 14 (20.29%) |
| **PPE ownership and use** | |
| Use PPE | 55 (79.71%) |
| Trained for use of PPE[a] | 63 (91.30%) |
| Own PPE | 50 (72.46%) |
| Never had any safety training at work | 30 (43.48%) |
| **Workplace safety environment** | |
| Eye PPE policy at workplace | 39 (56.52%) |
| **Protective eyewear use at time of injury** | |
| Welders wearing protective eyewear at time of injury | 56 (81.16%) |

N (%) indicates number of participants and percentage of the total; **Mean±SD** indicates the average and standard deviation; **SEE**, Secondary education examination; **SLC**, School leaving certificate; **PPE,** Personal protective equipment.

[a] Frequent Safe Training refers to safety education or demonstrations on the use of protective devices conducted at least once every six months, whereas Sometimes Safe Training refers to irregular or occasional safety instruction provided less than once per year.

## Workplace safety and training

Among all participants, 92.79% were trained in PPE use. An eye PPE policy was reported to be present in 57.66% of workplaces. Regarding safety training at work: 33.33% received it frequently, 32.43% sometimes, and 34.23% never received any formal safety training (Table 5).

**Table 5. Workplace factors affecting ocular trauma among welder (N = 111).**

| Characteristics | N(%) |
| --- | --- |
| **Trained for PPE use** | 103 (92.79%) |
| **Eye PPE policy at workplace** | 64 (57.66%) |
| **Safety training at work** | |
| Frequently | 37 (33.33%) |
| Sometime | 36 (32.43%) |
| Never | 38 (34.23%) |

N (%) indicates number of participants and percentage of the total; **PPE,** Personal protective equipment.

## Ocular disorders and visual acuity

Refractive errors were found in 9.91% of participants (myopia: 7.21%, hypermetropia: 1.80%, astigmatism: 0.90%). Presbyopia was present in 25.23%. On examination of 222 eyes, the most common findings were corneal opacity (20.72%), conjunctival congestion (10.36%), pinguecula (9.01%), and pterygium (3.60%). Cataract was found in 0.90% of eyes, and pseudophakia in 0.45%. Best-corrected visual acuity (BCVA) was 6/6 in 91% of right eyes and 92% of left eyes. One participant had perception of light only in one eye (Table 6).

## Spectacle use

A total of 17 welders (15.32%) used spectacles for distance vision, while only 9 (8.11%) used them while welding.

## Logistic regression analysis

Multivariate logistic regression revealed that male gender and lack of formal safety training were significantly associated with an increased risk of ocular trauma ($p < 0.05$) (Fig 4). In a separate model analyzing categorized age and welding hours per year, participants aged 46–60 years had significantly higher odds of reporting ocular trauma ($p < 0.05$) (Fig 5).

## Discussion

This study provides critical insights into the prevalence and risk factors for ocular trauma among welders in Hetauda, Nepal. The findings demonstrate a significantly high prevalence of ocular trauma (62.16%) among welders, highlighting the urgent need for occupational health interventions in Nepal's growing industrial sectors.

### Prevalence and patterns of ocular trauma

The reported ocular trauma prevalence in this study aligns with findings from similar settings. In previous studies done in Nepal, reveal a high incidence of work and welding-related ocular injuries, mirroring trends observed in other industrializing developing nations [8]. In a study done previously by Gurung et al, the prevalence of ocular trauma was 32.83%, while other studies showed lower incidences of 1.8% in Bhaktapur Eye Study [8,9]. In other studies, it ranged from 1.74% to 2.4% in Western Nepal and Nepal blindness survey respectively [10,11].

**Table 6. Ocular disorders among welders (N = 111).**

| Characteristics | N (%) |
|---|---|
| Refractive error[a] | 11 (9.91%) |
| Myopia | 8 (7.21%) |
| Hypermetropia | 2 (1.80%) |
| Astigmatism | 1 (0.90%) |
| Presbyopia | 28 (25.23%) |
| Examination finding[b] | |
| Conjunctival congestion | 23 (10.36%) |
| Pinguecula | 20 (9.01%) |
| Pterygium | 8 (3.60%) |
| Corneal opacity | 46 (20.72%) |
| Cataract | 2 (0.90%) |
| Pseudophakia | 1 (0.45%) |

**N (%)** indicates number of participants and percentage of the total.

[a]111 participants. Subcategories are non-exclusive (participants may have >1 condition).

[b]222 eyes.

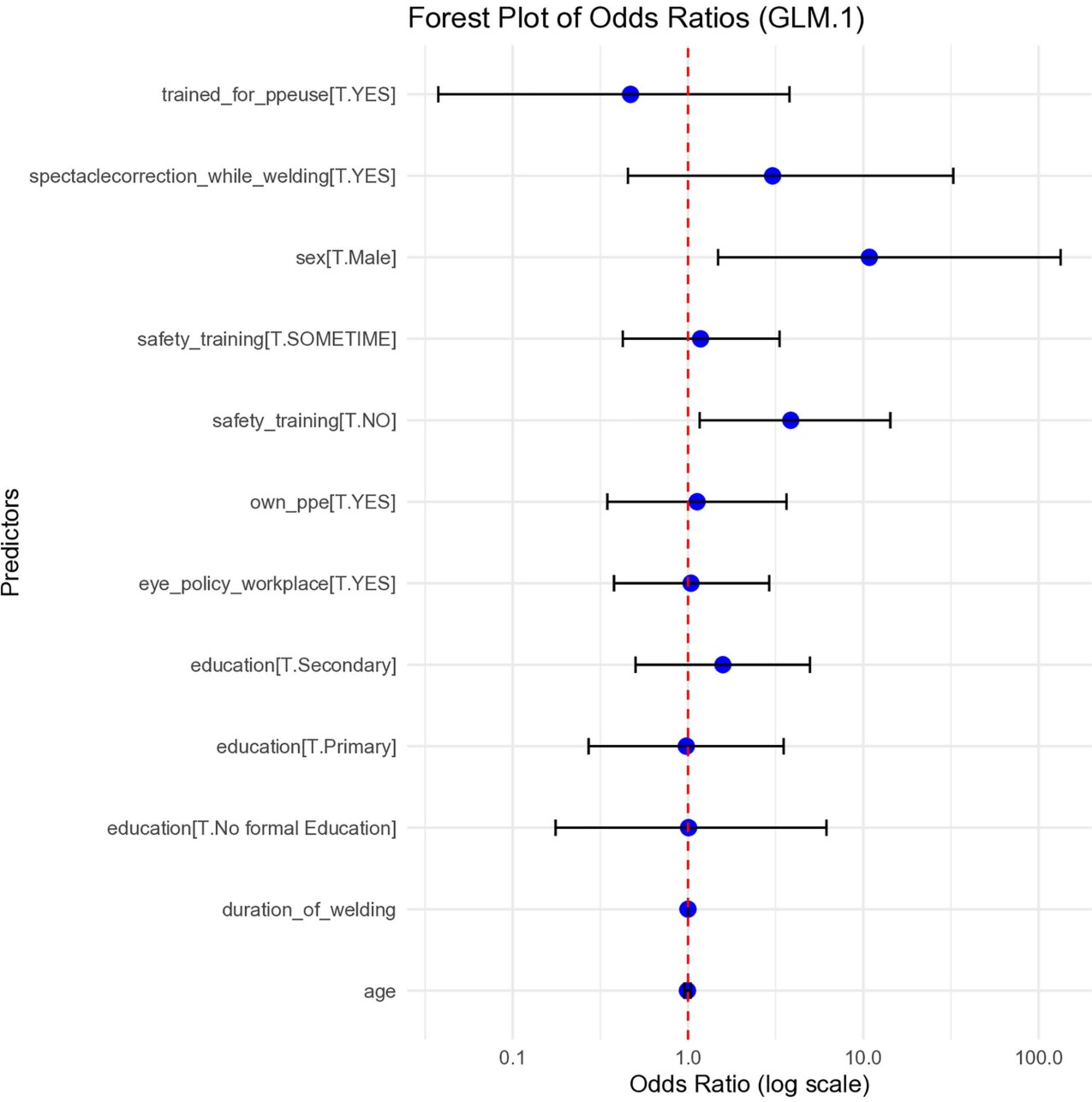

**Fig 4. Logistic regression analysis showing factors associated with ocular injury among welders.** The analysis indicates that male gender and lack of safety training were significantly associated with higher odds of ocular injury (p < 0.05). Odds ratios and 95% confidence intervals are shown for each factor.

Our rate (62.16%) is comparable to that found in studies conducted in Ghana and Nigeria [1,7], underscoring the occupational vulnerability of welders in low- and middle-income countries. The prevalence of ocular trauma in our cohort is also consistent with global findings, which report foreign body injuries, burns, and abrasions as the most common welding-related ocular injuries [1,3,4,12].

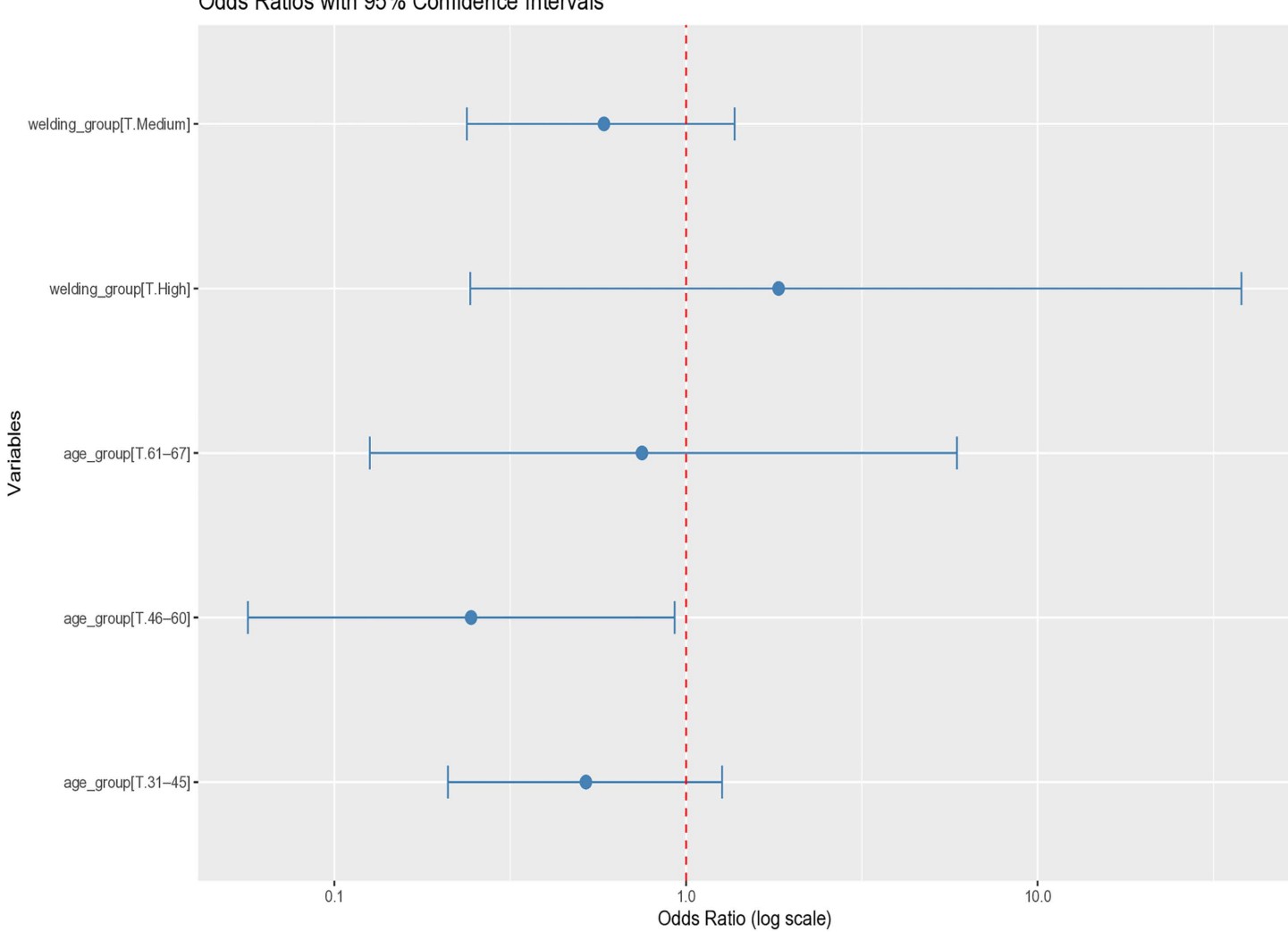

**Fig 5. Forest plot showing adjusted odds ratios for the association of age and welding duration with ocular trauma.** The plot shows adjusted odds ratios (ORs) with 95% confidence intervals for different age groups and welding duration. The red dashed line indicates the null value (OR = 1.0). Only the 46–60 age group demonstrated a statistically significant association with ocular trauma.

Interestingly, a large proportion of affected welders (79.71%) reported wearing some form of protective eye-wear at the time of injury. This raises concerns about the adequacy and effectiveness of the PPE used. Many welders may rely on non-standard or uncertified eyewear, which may not provide sufficient protection against the hazards they face. These findings are in line with studies in Indian and Nigeria [2,6,13], which reported that although PPE ownership was relatively high, the protective value was often compromised by poor design or incorrect usage.

While some international studies show a gradual decline in incidence due to improved safety protocols and enforcement [4], Nepal still demonstrates higher rates, likely reflecting limited awareness, inadequate access to protective devices, and weak occupational safety regulations [8,9].

### Sex distribution

The overwhelming male predominance in our study reflects the gendered nature of welding as an occupation in Nepal. Men are disproportionately employed in high-risk occupations such as welding, construction, mining, and metalwork, which inherently involve greater exposure to mechanical, thermal, and radiant hazards [8,14–17].

### PPE use and barriers

Despite 72.07% of participants owning PPE and 78.39% reporting PPE usage during work, a significant portion still experienced trauma. In a cross-sectional interview based prospective study, Ben et al reported 38.30% of the workers experienced some form of ocular injury and 68.3% never wore safety gear at work [16]. Some other stories also support this where low rates of PPE use can be documented [18,19]. Contradicting this, 78.39% in our study were wearing safety gear which may be because of the high risk job.

Barriers such as perceived low risk, discomfort, and lack of regulatory enforcement were commonly cited, echoing findings from other occupational studies. Notably, more than 60% of non-users cited the absence of a workplace mandate as a reason for not using eye protection. These gaps suggest that merely providing PPE is insufficient without accompanying behavioral reinforcement and systemic enforcement. The implementation of a well-communicated eye safety policy, combined with regular vision screenings and spot checks, plays a critical role in behavioral enforcement of workplace safety [3]. PPE use reinforce by employer can increase its use in the work place and decreases the risk of injuries [19]. Studies of Nils bull have shown that there is drastic reduction of incidence of ocular injury among metal workers after eye protection became mandatory [20]. The presence of workplace safety regulations was found to significantly enhance workers' awareness of occupational hazards [5]. In our study, 56.52% had eye PPE policy at workplace but 43.48%never had any safety training at work which shows a huge gap of improvement proper law enforcement.

Most of them used sunglasses as PPE (72.97%) followed by eye shield and protective goggles. Similar to our case, in a study by Budhathoki et al, sunglasses were the most used form of PPE, (74.3%). Though sunglasses are cheap and easily available, it is not among the recommended PPE [14]. The welding goggle with filter shade number 10 is recommended for eye protection, particularly in industry settings for heavy-duty welding work [21] In a study by Yetunde et al, Sabitu et al and Isahand Okojie,the most common PPE worn in other studies was welding goggles [13,15,22]. Welding goggles/face-shield use was seen in18% welders study by Budhathoki et al. [14].

The cost of PPE was a major barrier to consistent use, with many welders opting for cheap, substandard, or no protection. This likely contributes to the high prevalence of ocular trauma, as inadequate PPE leaves workers exposed to sparks, fragments, and radiation. Subsidized PPE or employer mandates may improve adherence and reduce injury rates [1,3,23].

### Training, workplace policy, and risk factors

Our study found that while 92.79% of welders reported being trained in PPE use, 34.23% had never received formal safety training at work. Logistic regression analysis confirmed that lack of formal safety training was significantly associated with ocular trauma. This reinforces the importance of not only PPE distribution but also structured and recurring safety education. Moreover, welders aged 46–60 years were found to be at higher risk for ocular trauma. This may reflect cumulative exposure or decreased compliance with protective practices over time.

In our study, higher proportion of male were prone to have ocular injury which is similar to other studies [24–29]. The maximum numbers of workers were from working age group as in other similar studies [3,16,24,30].

Among those who sustained ocular trauma, 91.30% had received training on the use of personal protective equipment (PPE), yet only 79.71% reported actual usage. This pattern aligns with findings from a study in Nigeria, where 98% of welders were aware of the risk of eye injury during welding, but only 15.3% were using protective eyewear at the time of

injury [7]. Similarly, Budhathoki et al. reported that 90% of welders were aware of PPE; however, only about half of them consistently used it [14]. In contrast, our study demonstrates a comparatively higher rate of PPE use among those aware and trained, though gaps between knowledge and practice still remain evident.

### Ocular morbidity and spectacle use

Apart from trauma, the prevalence of ocular disorders such as corneal opacity (20.72%) and conjunctival congestion (10.36%) suggests chronic exposure to harmful radiation and particulate matter. While 15.32% of welders used spectacles for distance vision, only 8.11% used them during welding. This disparity points to another critical barrier—lack of awareness or access to prescription safety eyewear that meets both refractive and protective needs.

### Policy implications

These findings highlight the gap between policy and practice. While Nepal's Labour Act mandates the use of safety devices in hazardous jobs, implementation in informal or small-scale workshops appears insufficient. Many welders use locally available, non-standard protective gear. There is a need for quality assurance, subsidies for certified PPE, and active enforcement of safety policies. Additionally, incorporating basic occupational eye health education into technical training curricula for welders could create lasting behavioral change.

## Conclusion

This study demonstrates a high prevalence of ocular trauma among welders in Hetauda, Nepal, with foreign body injuries, burns, and abrasions being the most common. Despite relatively high reported use of personal protective equipment (PPE), economic constraints, substandard equipment, and improper usage significantly compromise protection.

The findings highlight the urgent need for targeted occupational health interventions in small-scale industries in Nepal, including subsidized or employer-provided PPE and regular safety training programs. They contribute to the global understanding of welding-related ocular hazards, showing that low- and middle-income settings continue to experience disproportionately high injury rates despite widespread awareness of protective measures. The study also underscores the importance of systematic surveillance and reporting of occupational injuries to inform policy and improve preventive strategies.

Future research should include longitudinal or cohort studies to assess the incidence, recurrence, and long-term consequences of ocular trauma among welders. Such studies could also evaluate the effectiveness of interventions aimed at increasing PPE adherence and reducing occupational injuries. Overall, these findings emphasize the need for comprehensive occupational safety policies tailored to local contexts to reduce preventable eye injuries and improve the health and productivity of Nepal's welding workforce.

## Limitations

This study is subject to certain limitations. As a cross-sectional design, it cannot establish causal relationships. The reliance on self-reported data may have introduced recall bias, especially in reporting past ocular injuries and PPE use. Additionally, the use of a convenience sample limits the generalizability of the findings beyond the study population. The number of years of welding experience was not recorded in the proforma, which restricted analysis of the relationship between duration of exposure and ocular morbidity. Although fundoscopy and IOP assessment were performed for all participants, subtle posterior segment abnormalities may have been missed. The absence of long-term follow-up further restricts the ability to evaluate recurrent or chronic sequelae of welding-related ocular trauma.

## Supporting information

**S1 Data. Raw dataset used for analysis in this study.** The file contains anonymized participant-level data.
(XLSX)

## Acknowledgments

The authors would extend their deepest appreciation to all the study participants for their time and cooperation.

## Author contributions

**Conceptualization:** Sunil Thakali, Hom Bahadur Gurung.

**Formal analysis:** Sunil Thakali, Hom Bahadur Gurung.

**Investigation:** Mohini Shrestha, Dikshya Bista, Hom Bahadur Gurung.

**Supervision:** Dikshya Bista, Hom Bahadur Gurung.

**Writing – original draft:** Sunil Thakali.

**Writing – review & editing:** Aleena Gauchan, Dikshya Bista, Hom Bahadur Gurung.

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
