## [Decision Letter · Decision Letter 0]

16 Sep 2025

Dear Dr. Thakali,

Thank you for submitting your manuscript to PLOS ONE. After careful consideration, we feel that it has merit but does not fully meet PLOS ONE’s publication criteria as it currently stands. Therefore, we invite you to submit a revised version of the manuscript that addresses the points raised during the review process.

We look forward to receiving your revised manuscript.

Kind regards,

Ilker Kacer, Assoc. Prof. M.D.

Academic Editor

PLOS ONE

Journal Requirements:

3. Please ensure that you refer to Figure 3 in your text as, if accepted, production will need this reference to link the reader to the figure.

Reviewers' comments:

Reviewer's Responses to Questions

**Comments to the Author**

1. Is the manuscript technically sound, and do the data support the conclusions?

Reviewer #1: Yes

Reviewer #2: Partly

Reviewer #3: Yes

Reviewer #4: Yes

2. Has the statistical analysis been performed appropriately and rigorously?

Reviewer #1: Yes

Reviewer #2: No

Reviewer #3: Yes

Reviewer #4: Yes

3. Have the authors made all data underlying the findings in their manuscript fully available?

Reviewer #1: Yes

Reviewer #2: Yes

Reviewer #3: Yes

Reviewer #4: No

4. Is the manuscript presented in an intelligible fashion and written in standard English?

Reviewer #1: Yes

Reviewer #2: Yes

Reviewer #3: Yes

Reviewer #4: Yes

Reviewer #1: The authors present an interesting article regarding ocular trauma among welders in Hetauda, Nepal. The paper discusses an important issue that affects the lives of working welders. However; I suppose the following comments would improve it:

Introduction

- The use of PPE abbreviation is inappropriate, please revise and use the unabbreviated term only in the first instance.

- Please strengthen your introduction by reviewing similar articles from around the world. For example (https://pubmed.ncbi.nlm.nih.gov/35101659/ and https://pubmed.ncbi.nlm.nih.gov/15933411/ )

Methods

- In study design, could you provide sample size calculations, to put into perspective the representativeness of your sample.

- The inclusion criteria are not clear. Were all participants examined? Or only ones with ocular complaints? Please elaborate.

- “The questionnaire was adapted from a previous study conducted in Accra, Ghana, with minor modifications.” Please cite the study and explain the modifications with justification.

- How did you define ocular trauma? Clearly describe what was perceived as ocular trauma

Results

- Table 3 needs revision. I suppose (Own PPE) should be under (PPE ownership and use).

- Trained and untrained should be following each other not separated.

Discussion

- How does the Nepali statistics regarding ocular trauma among welders compare to global statistics? And how is this relationship justified?

- Male predominance in the results can be attributed to the nature of the job itself, welder females are much less than males, please elaborate how this fact can affect the interpretation of your results regarding sex-oriented findings.

- Cost of equipment was mentioned briefly. How does this impact adherence of welders to the use of PPEs?

- To the limitations, please add that long-term follow-up and cohort studies are required to allow for a deeper understanding of this issue.

- Most cited papers are on the older. Please use more recent references.

- Figures are of low quality.

Reviewer #2: Dear Authors,

Thank you for your valuable and applicable research in the field of health and safety. After careful consideration, I would like to make the following detailed and scientifically sound recommendations for improving the paper:

Abstract: The abstract should follow the usual structured format of scientific papers. It should clearly include separate sections for the introduction, methods, key findings, and conclusions. This structure enhances clarity and allows readers to quickly understand the scope and results of the study.

Keywords: Please use the PubMed term mesh to add to the list of keywords to better reflect the main topic of your study and improve indexing.

Introduction

The introduction should begin with references to international statistics and data relevant to the topic, providing a global context.

Then, discuss the specific harms and economic burdens related to the health area you are investigating.

Highlight the gap in research in the specific field of Nepal to clearly justify the necessity of your study.

A well-organized paragraph explaining the overall purpose and significance of the research should be included to establish logic and coherence.

Study Methodology

Ethical considerations related to your research should be explicitly stated in the text of the paper. This includes approval from relevant ethics committees and informed consent procedures. These statements are usually placed at the end of the paper but are essential.

Provide a detailed description of the study design and sample selection process. Specify whether the samples were selected randomly or by convenience (accessibility sampling).

Clearly state the exact type of study conducted (e.g., cross-sectional, cohort, experimental).

Refer to the questionnaire used in the study. If any changes were made to the original instrument, please justify these changes and report re-evaluated reliability and validity coefficients.

Including a table summarizing the components of the questionnaire along with the number of questions in each section will help readers better understand the data collection tool.

The data analysis method should be clearly explained. Describe the software used, the specific statistical tests applied, and the rationale for choosing those tests.

Conclusions

The conclusions section requires a more comprehensive interpretation of the results. Rather than a summary, it should provide a detailed explanation of the implications of the findings for existing knowledge and practice.

Addressing these points with scientific rigor and scrutiny will significantly improve the quality and clarity of your paper and ensure that it meets the standards of scholarly publication.

Best regards,

Reviewer #3: Good study

Concerns:

1. It is good if we could know the total population of the welders in the association that you contacted, this will help us to know the response rate for this study. That population would have helped you calculate an ideal sample size for this study. This will help us to interpret the findings of this study better at least for this locality.

2. The number of years/months each had spend in this vocation was completely absent, if this data was collected kindly analysis to see its association with ocular trauma.

3.What proportion of the female gender in this study has ocular trauma? it is important to know this, it will truly let us know if gender is really a factor associated with ocular trauma.

4. As said in the limitation; this study depends on self recall of ocular trauma which is highly subjecive, hence the finding cannot be generalized.

Reviewer #4: I would like to thank the authors for their work

However this manuscript does not add to the literature addressed prevalence and risk factors of ocular trauma among welders in Nepal with no new novel findings

**Do you want your identity to be public for this peer review?** For information about this choice, including consent withdrawal, please see our Privacy Policy

Reviewer #1: No

Reviewer #2: No

Reviewer #3: **Yes: ** Akinsola Aina

Reviewer #4: No

---

## [Author Response · Author response to Decision Letter 1]

12 Oct 2025

We have prepared and uploaded all the required files as instructed:

Response to Reviewers: a detailed, point-by-point rebuttal addressing all reviewer and editor comments.

Revised Manuscript with Track Changes: showing all edits made in response to the comments.

Manuscript: the clean, unmarked version of the revised paper.

---

## [Decision Letter · Decision Letter 1]

9 Nov 2025

Dear Dr. Thakali,

Thank you for submitting your manuscript to PLOS ONE. After careful consideration, we feel that it has merit but does not fully meet PLOS ONE’s publication criteria as it currently stands. Therefore, we invite you to submit a revised version of the manuscript that addresses the points raised during the review process.

We look forward to receiving your revised manuscript.

Kind regards,

Ilker Kacer, Assoc. Prof. M.D.

Academic Editor

PLOS ONE

Journal Requirements:

Reviewers' comments:

Reviewer's Responses to Questions

**Comments to the Author**

Reviewer #2: (No Response)

Reviewer #3: All comments have been addressed

2. Is the manuscript technically sound, and do the data support the conclusions?

Reviewer #2: Yes

Reviewer #3: Yes

3. Has the statistical analysis been performed appropriately and rigorously?

Reviewer #2: No

Reviewer #3: Yes

4. Have the authors made all data underlying the findings in their manuscript fully available?

Reviewer #2: Yes

Reviewer #3: Yes

5. Is the manuscript presented in an intelligible fashion and written in standard English?

Reviewer #2: Yes

Reviewer #3: Yes

Reviewer #2: Dear authors,

While thanking you for the corrections made in this article regarding the questionnaire, due to the changes in the questions, the validity of the questionnaire has not been explained. What is meant by the changes is unclear.

Due to the changes in the references, the changes are not clear.

Reviewer #3: Welldone for the good research work.

Please, kindly mention the number of all the registered welders that were initially invited for the visual screening out of which 111 welders consented to present themselves for the study. This will give us the response rate and further help us to interpret the findings of this study better as regards the whole welders in Nepal.

Kindly define Frequent Safe training and Sometimes safe training at work mentioned in Table 4.

Was fundoscopy done for any of this patient? No posterior segment ocular morbidity was mentioned.

Was intraocular pressure done for any of this patient?

Maculopathy is an important ocular morbidity in welders.

Please include the number of years each participants had been involved in welding .... this may be a factor that might help when instituting intervention.

**Do you want your identity to be public for this peer review?** For information about this choice, including consent withdrawal, please see our Privacy Policy

Reviewer #2: No

Reviewer #3: **Yes: ** Akinsola Aina

---

## [Author Response · Author response to Decision Letter 2]

17 Nov 2025

Dear Editor and Reviewers,

Manuscript title: Prevalence of Ocular Trauma and Barriers to Use of Personal Protective Devices Among Welders in Hetauda, Nepal

Authors: Sunil Thakali, Mohini Shrestha, Aleena Gauchan, Dikshya Bista, Hom Bahadur Gurung

We sincerely thank the Academic Editor and both reviewers for their valuable time, constructive feedback, and insightful comments that have helped us improve the quality and clarity of our manuscript. We have carefully revised the paper in response to each point raised. A detailed, point-by-point response follows.

Response to Reviewer #2

Comment 1: While thanking you for the corrections made in this article regarding the questionnaire, due to the changes in the questions, the validity of the questionnaire has not been explained. What is meant by the changes is unclear. Due to the changes in the references, the changes are not clear.

Response: Thank you for your valuable comment. The questionnaire used in this study was adapted from a validated study conducted among welders in Accra, Ghana [1]. Minor modifications were made to ensure cultural and occupational relevance for Nepal, including adjustments to occupational history questions to reflect local work practices, inclusion of Nepali terms for personal protective equipment, and adaptation of items to match the types of welding equipment commonly used in Hetauda workshops. These modifications were reviewed by occupational health and ophthalmology experts to maintain content validity. Additionally, the adapted questionnaire was pilot-tested among a small group of welders (n = 10) to ensure clarity, comprehension, and contextual appropriateness before full-scale data collection.

Response to Reviewer #3

Comment 1: Please, kindly mention the number of all the registered welders that were initially invited for the visual screening out of which 111 welders consented to present themselves for the study.

Response: All welders invited (n = 111) consented and participated in the study, giving a response rate of 100%. This has been clearly mentioned in the Results section.

Comment 2: Kindly define Frequent Safe Training and Sometimes Safe Training at work mentioned in Table 4.

Response: These terms have now been defined in a footnote of Table 4 as follows:

• Frequent Safe Training: Conducted regularly (at least once per year) with structured safety instructions on PPE use.

• Sometimes Safe Training: Conducted irregularly or informally, without consistent follow-up or formal structure.

Comment 3: Was fundoscopy done for any of the patients? No posterior segment ocular morbidity was mentioned. Was intraocular pressure done for any of the patients?

Response: Yes, fundoscopy was performed using a 90D lens under slit-lamp examination; no posterior segment abnormalities were found. Intraocular pressure (IOP) was measured for all participants using a non-contact tonometer (iCARE). These details have been added to the Methods section under Ophthalmic Examination.

Comment 4: Please include the number of years each participant had been involved in welding.

Response: We acknowledge that the number of years of welding experience was not recorded in our study proforma. This has now been added as a limitation in the Discussion section, and we recommend including it in future research to better assess its relationship with ocular morbidity.

Comment 5: Maculopathy is an important ocular morbidity in welders.

Response: Thank you for the observation. Fundoscopic examination did not reveal any cases of maculopathy among participants. This information has been clarified in the Results section.

---

## [Decision Letter · Decision Letter 2]

7 Dec 2025

Prevalence of Ocular Trauma and Barriers to Use of Personal Protective Devices Among Welders in Hetauda, Nepal

PONE-D-25-37148R2

Dear Dr. Thakali,

We’re pleased to inform you that your manuscript has been judged scientifically suitable for publication and will be formally accepted for publication once it meets all outstanding technical requirements.

Kind regards,

Ilker Kacer, Assoc. Prof. M.D.

Academic Editor

PLOS One

Additional Editor Comments (optional):

Reviewers' comments:

Reviewer's Responses to Questions

**Comments to the Author**

Reviewer #2: All comments have been addressed

Reviewer #3: All comments have been addressed

2. Is the manuscript technically sound, and do the data support the conclusions?

Reviewer #2: Yes

Reviewer #3: Yes

3. Has the statistical analysis been performed appropriately and rigorously?

Reviewer #2: Yes

Reviewer #3: Yes

4. Have the authors made all data underlying the findings in their manuscript fully available?

Reviewer #2: Yes

Reviewer #3: Yes

5. Is the manuscript presented in an intelligible fashion and written in standard English?

Reviewer #2: Yes

Reviewer #3: Yes

Reviewer #2: (No Response)

Reviewer #3: Authors have addressed all my comments.

Findings will be a tool of advocacy and health education for use of protective eye wear.

**Do you want your identity to be public for this peer review?** For information about this choice, including consent withdrawal, please see our Privacy Policy

Reviewer #2: No

Reviewer #3: **Yes: ** Akinsola Sunday Aina

---

## [Editor Report · Acceptance letter]

PONE-D-25-37148R2

PLOS One

Dear Dr. Thakali,

I'm pleased to inform you that your manuscript has been deemed suitable for publication in PLOS One. Congratulations! Your manuscript is now being handed over to our production team.

Kind regards,

on behalf of

Mr. Ilker Kacer

Academic Editor

PLOS One